# Metabolome and Microbiome Analysis to Study the Flavor of Summer Black Tea Improved by Stuck Fermentation

**DOI:** 10.3390/foods12183414

**Published:** 2023-09-13

**Authors:** Lianghua Wen, Lingli Sun, Ruohong Chen, Qiuhua Li, Xingfei Lai, Junxi Cao, Zhaoxiang Lai, Zhenbiao Zhang, Qian Li, Guang Song, Shili Sun, Fanrong Cao

**Affiliations:** 1College of Horticulture, South China Agricultural University, Guangzhou 510000, China; tea15779451015@163.com; 2Tea Research Institute, Guangdong Academy of Agricultural Sciences, Guangdong Provincial Key Laboratory of Tea Plant Resources Innovation & Utilization, Guangzhou 510640, China; sunlingli@tea.gdaas.cn (L.S.); chenruohong@tea.gdaas.cn (R.C.); liqiuhua@tea.gdaas.cn (Q.L.); laixingfei@tea.gdaas.cn (X.L.); caojunxi@tea.gdaas.cn (J.C.); laizhaoxiang@tea.gdaas.cn (Z.L.); zhangzhenbiao@tea.gdaas.cn (Z.Z.); 3Guangdong Academy of Agricultural Sciences, Sericultural & Agri-Food Research Institute, Key Laboratory of Functional Foods, Ministry of Agriculture and Rural Affairs, Guangdong Key Laboratory of Agricultural Products Processing, Guangzhou 510610, China; liq@gdaas.cn; 4Guangzhou Yitang Biotechnology Co., Ltd., Guangzhou 510277, China; gzytsw@126.com

**Keywords:** summer black tea, tea quality, metabolic profile, microbial community

## Abstract

Tea is the most popular and widely consumed beverage worldwide, especially black tea. Summer tea has a bitter and astringent taste and low aroma compared to spring tea due to the higher content of polyphenols and lower content of amino acids. Microbial fermentation is routinely used to improve the flavor of various foods. This study analyzed the relationship between the quality of black tea, metabolic characteristics, and microbial communities after microbial stuck fermentation in summer black tea. Stuck fermentation decreased the bitterness, astringency sourness, and freshness, and increased the sweetness, mellowness, and smoothness of summer black tea. The aroma also changed from sweet and floral to fungal, with a significant improvement in overall quality. Metabolomics analysis revealed significant changes in 551 non-volatile and 345 volatile metabolites after fermentation. The contents of compounds with bitter and astringent taste were decreased. Sweet flavor saccharides and aromatic lipids, and acetophenone and isophorone that impart fungal aroma showed a marked increase. These changes are the result of microbial activities, especially the secretion of extracellular enzymes. *Aspergillus*, *Pullululanibacillus*, and *Bacillus* contribute to the reduction of bitterness and astringency in summer black teas after stuck fermentation, and Paenibacillus and *Basidiomycota_gen_Incertae_sedis* contribute positively to sweetness. In addition, Aspergillus was associated with the formation of fungal aroma. In summary, our research will provide a suitable method for the improvement of tea quality and utilization of summer tea, as well as provide a reference for innovation and improvement in the food industry.

## 1. Introduction

Tea is second in popularity among beverages in the world due to its unique flavor and potential health benefits such as antioxidant, anti-inflammatory, immunomodulatory, anti-cancer, cardiovascular protection, anti-fat, and weight loss [1,2]. Tea is divided into three categories: spring tea, summer tea, and autumn tea. The greatest tea is spring tea since it has a smooth, refreshing taste and an intense scent. Summer tea, on the other hand, is of inferior quality because of its astringent and bitter flavor, as well as its shortage of scent. Autumn tea has a middle-ground quality compared to the other two types. Therefore, summer tea has low economic viability and involves considerable wastage of resources. Tea quality is assessed based on its fragrance, flavor, and color, all of which are influenced by the composition of volatile and non-volatile ingredients. Catechins, the main class of tea polyphenols, are responsible for the astringent and bitter taste of the tea soup, while flavonols, phenolic acids, caffeine, and organic acids impart acidity, bitterness, and astringency [3]. The major contributors in the tea soup that contribute to its freshness and sweetness are amino acids and saccharides [3]. Some amino acids produce aromatic volatile compounds through enzymatic reactions, Maillard reactions and Strecker degradation. For example, methionine produces a potato-like aroma through Strecker degradation [4].

Microbial fermentation is routinely performed to enhance and modify the taste of food and beverages. For instance, Wang et al. enhanced the aroma of coffee beans by fermenting them with Saccharomyces cerevisiae and *Pichia kluyveri* [5]. Furthermore, Xiao et al. showed that fermentation of autumn tea with *Eurotiumcristatum* not only increased the floral flavor but also reduced the astringency and greatly improved the quality [6]. Chen et al. were able to incorporate herbal, sweet, minty, and floral aromas into the tea by co-fermenting with *E. cristatum* and *As*pergillus niger [7]. In addition, the fermentation of steamed green tea by *A.* PW-2 resulted in a decrease in the content of green aromatic volatile compounds and a decrease in the content of metabolites with astringent flavors, which may be related to extracellular enzymes such as peroxidase and glycoside hydrolase [8]. An et al. also reported significant changes in the color and aroma of instant green tea powder after fermentation with *E. cristatum* [9]. Tea is separated into the categories of green, black, yellow, dark, white, and oolong depending on the processing technique. Dark tea is the most fermented type, and its quality is primarily determined by the microorganisms that enzymatically convert or metabolize tea substances into aromatic compounds [10]. Black tea is the most consumed tea in the world. Compared to spring black tea, summer tea has poor quality but high yield. Therefore, improving the quality of summer black tea is important economically. Only a few studies have reported the use of microorganisms to ferment black tea in order to improve its quality. Summer black tea was fermented using *Cordyceps militaris* by Zhang et al. [11]. They discovered that the level of total polyphenols and total flavonoids was reduced while the level of components with floral and woody scents rose [11].

Metabolomics methods such as UPLC-MS/MS and GC-MS/MS, which are characterized by excellent throughput, superior sensitivity, and broad coverage, have been widely used in the study of the composition of non-volatile and volatile metabolites in tea. In addition, ITS sequencing and 16S rRNA sequencing are routinely used to analyze the microbial composition of biological samples. The purpose of the present research was to assess the effects of microbial stuck fermentation on the quality of summer black tea in terms of the changes in taste and aroma. UPLC-MS/MS and GC-MS/MS were applied to examine the metabolic changes. Additionally, we compared the microbial communities of the fermented and unfermented tea samples using ITS and 16S rRNA sequencing, and we clarified the relationship between the microbial communities, tea quality, and metabolites. Our findings provide an experimental basis for improving summer tea quality through microbial fermentation.

## 2. Materials and Methods

### 2.1. Experimental Materials

The Tea Research Institute of Guangdong Academy of Agricultural Sciences (Yingde City, Guangdong Province, China) provided fresh tea for Yinghong No. 9, and it were chosen in accordance with plucking guidelines that called for a bud with two leaves. Merck provided the chromatographic grade methanol, acetonitrile, and hexane; Aladdin provided the chromatographic grade formic acid; and Sinopharm Chemical Reagent Co., Ltd. (Guangzhou city, Guangdong Procince, China). provided the analytical quality sodium chloride.

### 2.2. Tea Preparation

Tea leaves of Yinghong No. 9 are commonly processed into black tea, and the use of black tea as a base for microbial fermentation has rarely been reported, and generally microbial fermentation teas are based on green tea [12]. The approximate process of summer black tea (YH)is as follows: fresh leaves were naturally wilted to a moisture content of about 55% (25–27 °C, 65–70% relative moisture), and then rolled twice in a rolling machine for 45 min each time. For 8 h, the crushed leaves were fermented at 25–27 °C and 85% relative moisture in a bamboo basket. Finally, the leaves were dried at 120 °C for 20 min, and thereafter at 80 °C for 2 h. Stuck-fermented summer black tea (FHYH) was prepared by evenly spraying the YH with water to maintain the moisture content at 30–35%, followed by 35 days of stuck fermentation (22–24 °C, 70–75% relative moisture). The leaves were turned over every 7 days. The fermented tea leaves were dried to a moisture content of 4–5% (Appendix A). Six biological replicates were prepared for each tea. For further examination, the two tea samples were kept in liquid nitrogen.

### 2.3. Analysis of Sensory Evaluation

Eight experts (including four men and four women in the group), ranging in age from 25 to 60, who were not allowed to ingest irritating or spicy food before the sensory evaluation, assessed the tea samples. The YH and FHYH samples were assessed in accordance with the Chinese standard (GB/T 23776-2018) based on appearance, color, taste, and scent (100-point scale) [13], as well as quantitative descriptive analysis (QDA) on 12 taste and aroma attributes as 0 (none), 5 (mild), and 10 (strong). Each of the samples was evenly spaced over a white teaboard that was 23 cm × 23 cm × 3.3 cm in size. 3 g of YH or FHYH were steeped in 150 mL of boiling water for five minutes after taking note of the shape, color, integrity, and clarity. The brewed tea was then poured into a bowl. In addition to the taste, aroma, and color of the tea soup, the appearance of the leaf base was also noted. Sensory evaluation was carried out at 15–20 °C and relative humidity <70%.

### 2.4. Analysis of Non-Volatile Metabolites by UPLC-MS/MS

#### 2.4.1. Sample Preparation and Extraction

The YH or FHYH samples were vacuum freeze-dried (Scientz-100F, NINGBO SCIENTZ BIOTECHNOLOGY CO., LTD, Ninbo City, Zhejiang Province, China) and then ground with a zirconia bead for 1.5 min at 30 Hz in a mixer mill (MM 400, Retsch). 50 mg of each lyophilized powder was submerged in 1.2 mL of 70% methanol and vortexed for 30 s each time for a total of 6 times to create the extract. For the UPLC-MS/MS analysis, the extracts were centrifuged at 12,000 rpm for 3 min before being filtered (SCAA-104, 0.22 m pore size).

#### 2.4.2. UPLC Conditions

The YH and FHYH extracts were examined using UPLC-ESI-MS/MS (UPLC, SHIMADZU Nexera X2; MS, Applied Biosystems 4500 Q TRAP, Thermo Fisher Scientific, Shanghai City, China). For UPLC, an Agilent SB-C18 column (1.8 m, 2.1 mm × 100 mm) was employed. Both solvents contained 0.1% formic acid: solvent A, which was pure water, and solvent B (mobile phase), which was acetonitrile. The initial gradient for the sample analyses was 95% A and 5% B. Within 9 min, a gradient that was linear of 5% A and 95% B was established and kept constant for 1 min. Within 1.1 min, the gradient was altered to 95% A and 5% B, where it remained for 2.9 min. The flow velocity was 0.35 mL per minute, the injection volume was 4 μL, and the column oven was set to 40 °C. The effluent and an ESI-triple quadrupole-linear ion trap (QTRAP)-MS were connected differently.

#### 2.4.3. ESI-QTRAP-MS/MS

The ion spray voltage (IS) was 5500 V (positive ion mode)/−4500 V (negative ion mode) and the source temperature was 550 °C. The collision-activated dissociation (CAD) was high, and the ion source gases I (GSI), II (GSII), and curtain gas (CUR) were set to 50, 60, and 25 psi, respectively. Polypropylene glycol solutions containing 10 and 100 mol/L were utilized for instrument calibration and mass calibration for the QQQ and LIT modes, respectively. QQQ scans were obtained from MRM tests using colliding nitrogen gas. Individual MRM transitions’ declustering potential (DP) and collision energy (CE) were improved. Based on the metabolites eluted at each interval, a specific set of MRM transitions has been investigated for each period.

### 2.5. Analysis of Volatile Metabolites by GC-MS/MS

#### 2.5.1. Sample Preparation and Treatment

A 20 mL head-space vial (Agilent, Palo Alto, CA, USA) holding a saturated solution of NaCl was filled with 500 mg of each of the YH or FHYH samples after it had been ground into powder form. Crimp-top lids from Agilent with TFE-silicone headspace septa were used to close the vials. A 120 m DVB/CWR/PDMS fiber from Agilent was heated in each vial for 5 min at 60 °C before being exposed to the headspace for 15 min at 100 °C for the SPME investigation.

#### 2.5.2. GC-MS Conditions

Following the sampling process, the VOCs were absorbed from the fiber coating in the injector port of the GC device (Model 8890; Agilent) for 5 min while operating in splitless mode. An Agilent Model 8890 GC and an Agilent 7000D mass spectrometer outfitted with a 30 m × 0.25 mm × 25 cm DB-5MS (5% phenyl-polymethylsiloxane) capillaries columns were used to identify and measure VOCs. The medium that carried gas was helium, moving at a linear speed of 1.2 mL/min. The detection temperature was set at 280 °C, while the injection device was kept at 250 °C. The oven was preheated to 40 °C for 3.5 min, and then the temperature was raised at a rate of 10 °C per minute, 7 °C per minute, and 25 °C per minute until 280 °C, where it remained for 5 min. At an energy of 70 eV, mass spectra have been collected in the electron impact (EI) ionization modality. The quadrupole mass detection device, ion source, and transmission line were all adjusted to temperatures of 150 °C, 230 °C, and 280 °C, respectively. Analyte identification and quantification were performed using the MS ion monitoring (SIM) mode.

### 2.6. Analysis of Microbial Community by 16S rRNA and ITS Sequencing

Employing the CTAB technique, total genomic DNA was isolated from YH or FHYH, quantified, and the purity was checked on 1% agarose gels. The DNA samples were diluted to 1 ng/μL in sterile water, and used as templates for 16S and ITS metagenomic analysis using the Illumina HiSeq platform (Beijing Novogene Genomics Technology Co., Ltd., Beijing, China, Wuhan Metawell Biotechnology Co. Ltd., Wuhan, Hubei Province, China). The 16s rRNA library was prepared by amplifying the V4 h region of the 16S rRNA gene with 515F (GTGCCAGCMGCCGCGG) and 806R (GGACTACHVGGGTWTCTAAT) primers. The ITS library was generated by amplifying the target sequences ITS1 (CTTGGTCATTTAGAGGAAGTAA) and ITS2 (GCTGCGTTCTTCATCGATGC). All PCR reactions were performed using 15 μL Phusion^®^ High-Fidelity PCR Master Mix (New England Biolabs, Ipswich, MA, USA), 2 μM each of forward and reverse primers, and 10 ng template DNA. Using the TruSeq^®^ DNA PCR-Free Sample Preparation Kit (Illumina, San Diego, CA, USA), sequencing libraries were created, and index codes were added. The Qubit@ 2.0 Fluorometer (Thermo Fisher Scientific, Shanghai City, China) and the Agilent Bioanalyzer 2100 system (Agilent Technologies, Santa Clara, CA, USA) were used to evaluate the library’s quality. The Illumina NovaSeq platform was used to finish sequencing the collection. After that, FQtrim (V0.94) was utilized to obtain superior clean readings. With the aid of QIIME2, alpha, and beta diversity were computed in order to measure species richness and diversity. Principal component analysis (PCoA) was used to examine the changes in microbiological makeup. Linear discriminant analysis (LDA) with linear discriminant effect size (LEfSe) was used for species analysis with significant differences.

### 2.7. Statistical Analysis

The YH and FHYH samples were analyzed by UPLC-MS/MS in triplicate, and six replicates for each sample were used for GC-MS/MS,16S rRNA sequencing, and ITS sequencing. Applying R (http://www.r-project.org, (accessed on 23 March 2023)), hierarchical cluster analysis (HCA) and principal component analysis (PCA) were carried out after the levels of the metabolites were normalized (Unit variance scaling). While HCA was clustered using a complete linkage clustering technique, PCA and HCA were both computed using Euclidean distance. The data must first be log2 converted, followed by zero-centered normalization, before being analyzed using R (http://www.r-project.org, (accessed on 23 March 2023)). This is necessary for the orthogonal partial least square-discriminant analysis (OPLS-DA). The OmicStudio tool program (https://www.omicstudio.cn/tool, (accessed on 12 May 2023)) was used to generate heat maps of various non-volatile metabolites and for Spearman-based correlation analysis [14]. Utilizing the KEGG Compound database (http://www.kegg.jp/kegg/compound/, (accessed on 24 March 2023)), the discovered metabolites were annotated. The QDA was graphically presented and analyzed using Origin 2022 and IBM SPSS Statistics 26 respectively. In addition, FlavorDB (https://cosylab.iiitd.edu.in/flavordb/, RL (accessed on 15 April 2023)), Flavor Library (https://www.femaflavor.org/flavor-library, (accessed on 15 April 2023)), and Flavornet (http://www.flavornet.org/flavornet.html, (accessed on 15 April 2023)) were used to obtain the odor characteristics and aroma thresholds of volatile metabolites. The taste characteristics and thresholds of non-volatile metabolites were obtained via a literature review [15,16].

## 3. Results and Discussion

### 3.1. Stuck Fermentation Improved the Sensory Quality of Summer Black Tea

The flavor characteristics of YH and FHYH samples were determined through traditional sensory evaluation (Figure 1 and Appendix A). Compared to YH, the dry FHYH tea leaves appeared wirier with a yellowish auburn color, although there was no appreciable difference in the integrity and clarity of both samples (Figure 1A,E). The YH tea soup had a clear orange-red color, whereas the FHYH soup appeared deep red and dull (Figure 1B,F). The leaf base of YH was orange-red and that of FHYH was reddish brown, and both types showed soft and even leaf bases (Figure 1C,G). In terms of taste, YH was strongly bitter and astringent with a slight acidity, while FHYH was more mellow and smooth with a slight sweetness. However, YH had a fresher taste compared to FHYH (Figure 1D), which can be attributed to the conversion of bitter and astringent substances by microbial fermentation. Consistent with our hypothesis, Xiao et al. showed that stuck fermentation can significantly improve the quality of autumn tea [6]. It is interesting to note that Xiao et al. used a single strain (*E. cristatum*) for fermentation of green tea, whereas we employed a natural stuck fermentation involving numerous microorganisms, which further illustrates the important role of microbial fermentation in tea flavor improvement. The aroma of YH tea was strong, sweet, and floral aroma, while that of FHYH was weak and light. In addition, fungal aroma was more prominent and lasting in FHYH, whereas both types had similar green and woody aromas (Figure 1H). The overall quality of FHYH was higher compared to YH, with respective sensory scores of 87 and 93. Taken together, stuck fermentation decreased the bitterness, astringency, sourness, and freshness of summer black tea, and enhanced the sweetness, mellowness, and smoothness, along with altering the aroma profile from sweet and floral to fungal.

### 3.2. Stuck Fermentation Altered the Profile of Non-Volatile Metabolites in Summer Black Tea

We examined the metabolomes of YH and FHYH using UPLC-MS/MS in order to ascertain the impact of stuck fermentation on the non-volatile metabolites of summer black tea. Signal stability was shown by the substantial curve overlap of total ion flow in metabolites identification in both positive and negative ion modes (Appendix A). Additionally, more than 75% of the quality control (QC) samples included components with a coefficient of variation (CV) less than 0.3, indicating robust experimental results (Appendix A).

A total of 1248 non-volatile metabolites were detected, including 106 amino acids and derivatives, 85 alkaloids, 305 flavonoids, 42 lignans and coumarins, 162 lipids, 68 nucleotides and derivatives, 95 organic acids, 228 phenolic acids, 28 tannins, 14 terpenoids, and 115 other metabolites (Figure 2A and Appendix A). The YH and FHYH samples were split into two different clusters based on these metabolites, and the three repetitions of every group were grouped collectively and recognizable from one another. (Figure 2B). This indicated considerable homogeneity among the replicates, along with high data reliability. An et al. also reported that the metabolites of tea samples with different levels of microbial fermentation showed significant separation on PCA analysis [17]. Furthermore, HCA (Figure 2C) and OPLS-DA (Figure 2D) also confirmed the significant differences between YH and FHYH, demonstrating the dependability of the model and lack of overfitting (Appendix A). Significant alterations in these metabolites after stuck fermentation were also demonstrated in a study tracking the dynamic changes in nonvolatile metabolism during stuck fermentation of green tea, which is consistent with our research [18]. Thus, stuck fermentation significantly altered the metabolic profile of summer black tea.

We identified 551 differential non-volatile metabolites between the two groups (Figure 2E, Appendix A, VIP ≥ 1, *p* ≤ 0.05, |Log_2_FC| ≥ 1), and KEGG enrichment analysis showed that these metabolites were enriched in 14 pathways, including ABC transporters (ko02010), aminoacyl-tRNA biosynthesis (ko00970) and metabolic pathways (ko01100) (Figure 2F, *p* ≤ 0.05). Notably, 6 of the 14 enriched pathways are related to amino acids, including biosynthesis of amino acids (ko01230), histidine metabolism (ko00340), valine, leucine and isoleucine degradation (ko00280), alanine, aspartate and glutamate metabolism (ko00250), tryptophan metabolism (ko00380), phenylalanine, tyrosine and tryptophan biosynthesis (ko00400). These results suggest that stuck fermentation of summer black tea may affect the relative content of metabolites and amino acid metabolism through multiple pathways. The variations in amino acids and their derivatives, phenolic acids, flavonoids, organic acids, alkaloids, saccharides, and lipids—all of which are directly connected to the flavor of summer black tea—are investigated more thoroughly in the sections that follow.

#### 3.2.1. Amino Acids and Derivatives

The scent and flavor of the tea soup are benefited by the presence of free amino acids [19]. We identified 56 differential amino acids and derivatives between the YH and FHYH samples, of which 6 were significantly upregulated in FHYH, while the remaining 50 showed a decreasing trend (Figure 3A). These findings are consistent with the study of An, who reported a general decrease in the free amino acid content of instant tea powder after liquid-state fermentation [17]. Therefore, the decrease in amino acid concentration may be applied to clarify why FHYH is less fresh. Specifically, we detected a considerable decrease in L-glutamine (CAS:56-85-9), L-tyrosine methyl ester (CAS:1080-06-4), L-proline (CAS:147-85-3), L-isoleucine (CAS:73-32-5), L-tyrosine (CAS:60-18-4), L-tryptophan (CAS:73-22-3), L-histidine (CAS:71-00-1), L-leucine (CAS:61-90-5), L-valine (CAS:72-18-4), and L-phenylalanine (CAS:63-91-2). Previous studies have shown that tryptophan, L-tyrosine methyl ester, L-proline, L-isoleucine, and L-tyrosine are the determinants of the astringent and bitter taste of tea, while L-glutamine (CAS:56-85-9) impart a salty taste to tea soup [10,16]. Furthermore, several bitter amino acids (e.g., histidine, leucine, valine, and phenylalanine) can be attenuated by γ-glutamylization [20], which may be responsible for the reduced bitterness and astringency of FHYH. Similar results were published by Wang et al., who demonstrated that following microbial fermentation, tea had less L-glutamic acid, L-norleucine, L-phenylalanine, and L-tyrosine [21]. In addition, the glycoconjugated amino acids L-glutamine-O-glycoside and L-glutamic acid-O-glycoside, which impart a sweet taste to tea soup by releasing glycosides, also decreased after stuck fermentation. We also detected a significant accumulation of N-acetyl amino acids, such as N-acetyl-L-tryptophan (CAS:1218-34-4) and N-acetyl-L-glutamine (CAS:2490-97-3), in FHYH. Likewise, Chen et al. also found that several N-acetyl amino acids increased in tea after microbial fermentation, and predicted their potential neuroprotective effects by molecular docking [22]. To summarize, the decrease in the content of amino acids and their derivatives due to stuck fermentation alleviated the bitterness and astringency of tea soup, decreased the freshness, and increased sweetness.

#### 3.2.2. Phenolic Acids

Phenolic acids are beneficial bioactive compounds and the primary building blocks of flavonols and catechins. They are the determinants of the color and flavor of tea soup and the main source of bitterness and astringency [23]. Stuck fermentation led to a significant decrease in phenolic acids (Figure 3B), especially those connected to glycosides, such as 6-O-galloyl-β-D-glucose (CAS:13186-19-1) and protocatechuic acid-4-O-glucoside. Microorganisms contribute to tea quality by secreting enzymes that break these glycosidic linkages and release sweet substances that directly relieve the bitterness of phenolic acids [8]. Chlorogenic acid (CAS:327-97-9) and neochlorogenic acid (CAS:906-33-2) impart a bitter taste to the tea soup, and were reduced by microbial stuck fermentation. Similar results were reported by Wang et al. as well [21]. Several hydroxycinnamic acids also contribute to the flavor of tea soup. For instance, Wen et al. identified trans-4-O-p-coumaroylquinic acid as a low-threshold astringent compound in tea [24]. We observed a reduction in 4-O-p-coumaroylquinic acid levels (CAS:32451-86-8) after stuck fermentation, which may partly explain the obvious reduction in the bitterness and astringency of the fermented summer black tea. Some phenolic acids showed an increase after stuck fermentation, including caftaric acid (CAS:67879-58-7), 3-(4-Hydroxyphenyl)-propionic acid (CAS:501-97-3), and syringic acid (CAS:530-57-4). Yu et al. also observed an increase in these phenolic acids in the early stages of fermentation [23]. Overall, the improvement in bitterness and astringency of summer black tea may be related to the reduction in certain phenolic acids with this flavor profile.

#### 3.2.3. Flavonoids

Flavonoids, which are also accountable for the color and taste of tea soup, particularly the astringency and bitterness, may have health advantages [21]. In addition, flavonoids also intensify the bitterness of other substances, such as coffee [23]. We screened 73 flavonoids that were significantly altered after stuck fermentation, of which 53 showed a decreasing trend in FHYH (Figure 3C), including the strongly astringent catechin gallate (CAS:130405-40-2), epicatechin gallate (CAS:1257-08-5) and epigallocatechin-3-gallate (CAS:989-51-5). Consistent with our findings, Yu et al. reported that 113 flavonoids were down-regulated and only 20 flavonoids were up-regulated after microbial fermentation of tea [23]. Furthermore, flavonoid glycosides including thataromadendrin-7-O-glucoside (CAS:28189-90-4) and phloretin-2′-O-glucoside (phlorizin) (CAS:60-81-1) decreased after microbial fermentation, which was accompanied by a concomitant increase in their flavones, such as aromadendrin (dihydrokaempferol) (CAS:480-20-6) and phloretin (CAS:60-82-2). We hypothesize that the thermal and enzymatic hydrolysis of glycosides likely reduced the bitter taste of flavonoids [8], which is consistent with prior research [23]. Interestingly, while isosalipurposide (CAS:4547-85-7), quercetin-3-O-sophoroside-7-O-rhamnoside and naringenin-4′-O-glucosideshow also decreased during fermentation, there was no increase in the corresponding flavones, most likely since the latter may immediately combine with other compounds. Furthermore, all myricetin glycosidic compounds, including myricetin-3-O-galactoside-3′-O-rhamnoside and myricetin-3-O-(6″-malony) glucoside, decreased after stuck fermentation. In contrast, all myricetin-related other compounds such as myricetin-3-O-sulfonate and dihydromyricetin (ampelopsin) (CAS:480-20-6) showed an increase. After microbial stuck fermentation, the color of the black tea soup may become more intense due to the conversion of flavonoids to other substances with a dark hue. Li et al. also showed that microbial fermentation deepened the color of the tea soup, which was attributed to the increased levels of theabrownins, the oxidized polymers of catechins [8]. Together, these results indicate that the reduction of flavonoids caused by microbial stuck fermentation alleviates the bitterness and astringency of summer black tea, and intensifies the color of tea soup.

#### 3.2.4. Organic Acids

Organic acids are a class of sour substances that lend a disagreeable flavor to the tea. A total of 60 organic acids were significantly altered after fermentation (Figure 3D), of which citric acid (CAS:77-92-9), γ-aminobutyric acid (CAS:56-12-2), glutaric acid (CAS:110-94-1), 2-hydroxyisobutyric acid (CAS:594-61-6), isocitric acid (CAS:320-77-4), cis-aconitic acid (CAS:585-84-2), L-pipecolic Acid (CAS:3105-95-1), 6-aminocaproic acid (CAS:60-32-2) and succinic acid (CAS:110-15-6) were down-regulated. Citric acid, glutaric acid, cis-aconitic acid, isocitric acid, and succinic acid have an acidic taste, 2-hydroxyisobutyric acid, L-pipecolic acid, and 6-aminocaproic acid increase the bitterness of tea soup, and γ-aminobutyric acid brings a dry feeling to the mouth. In addition, citric acid also increases the bitterness of the tea soup by increasing the leaching rate of polyphenols [25]. The decrease in these metabolites may be one of the potential reasons for the mellowness and smoothness, and decreased bitterness and sourness of summer black tea after stuck fermentation. In addition, L-malic acid (CAS:97-67-6) was also significantly down-regulated in FHYH. While Chen et al. reported a significant increase in citric acid, succinic acid, and malic acid after traditional fermentation of black tea [26], another study showed that these metabolites were reduced in microbially fermented dark tea [23]. One possible explanation is that traditional fermentation of black tea involves oxidation via endogenous enzymes (e.g., polyphenol oxidase) in the tea leaves under humid and warm conditions, whereas stuck fermentation entails conversion of substances through microbial activity and secretion of extracellular enzymes [27]. Currently, our findings suggest that the changes in organic acids brought on by stuck fermentation ultimately reduce the bitterness and sourness of summer black tea and favor a more mellow and smooth flavor.

#### 3.2.5. Alkaloids

The alkaloids are ubiquitous bitter compounds in herbs and plant-derived beverages and are associated with euphoric effects. As shown in Figure 3E, 46 alkaloids were significantly altered in FHYH, of which 31 were down-regulated. Histidinol (CAS:501-28-0), piperidine (CAS:110-89-4), and O-phosphorylethanolamine (CAS:1071-23-4) showed a decreasing trend. While histidinol and piperidine have a bitter taste, O-phosphorylethanolamine contributes to acidity [16]. Theophylline (CAS:58-55-9), a common alkaloid present in tea along with caffeine and theobromine, was significantly increased in FHYH compared to YH. According to Zhou et al., theophylline concentration in tea rose during *Aspergillus sydowii* fermentation, which may be connected to the microbial secondary metabolism’s degradation of caffeine [28]. However, caffeine levels were similar in YH and FHYH, which requires further study. Taken together, the reduced content of 31 alkaloids may explain the lower bitterness and acidity of summer black tea after stuck fermentation.

#### 3.2.6. Saccharides

Studies show that the sweetness of tea depends on the presence of saccharides. As shown in Figure 3F, the content of D-sorbitol (CAS:50-70-4), ribitol (CAS:488-81-3), dulcitol (CAS:608-66-2), xylitol (CAS:87-99-0), L-arabitol (CAS:7643-75-6) and D-arabitol (CAS:488-82-4) increased after stuck fermentation. These sugar alcohols are widely used as sweeteners in the food industry due to their high sweetness and low glycemic index. In addition, we also detected a significant increase in sucrose (CAS:50-99-7) levels in FHYH. On the other hand, D-glucose (CAS:50-99-7) and D-fructose (CAS:57-48-7) levels decreased after fermentation, which may be due to the microbial synthesis of sucrose, or utilization of glucose and fructose by *Aspergillus* as a source of energy [29]. Ma et al. also reported a decrease in D-glucose and D-fructose in microbially fermented tea [30]. The concentration of sweet compounds in the tea soup, however, is below the threshold concentration for taste perception, according to earlier research, and they thus do not considerably add to the sweetness of the soup [31]. Nevertheless, we have evidence that stuck fermentation can increase the content of saccharides in summer black tea, ultimately enhancing the sweetness of the tea soup.

#### 3.2.7. Lipids

Lipids are essential precursors of the aromatic substances in tea [32]. After stuck fermentation, an overall of 60 lipids were elevated, whereas just seven were down-regulated (Figure 3G). Free fatty acids (FFAs), lyso-phosphatidylcholine (LPC), lyso-phosphatidylethanolamine (LPE), and glycerol ester were the four primary lipids that increased significantly in the fermented tea. Li et al. also reported an increase in FFAs, LPE, and LPC in tea soup after fermentation, and proposed that FFAs are likely formed during lipolytic reactions driven by microbial hydrolases [32]. Glycerophospholipids, which are crucial elements of the plasma membrane, degrade into LPE and LPC [20]. According to one research, microbes cover the surface of tea leaves after fermentation and drastically change their cell structure [33]. Therefore, we hypothesized that the elevated LPE and LPC in FHYH may be the result of the degradation of the microbial plasma membrane. Linoleic acid (CAS:60-33-3), a polyunsaturated fatty acid that was significantly up-regulated after stuck fermentation, is a precursor of an aliphatic aromatic substance that contributes to specific aromas in microbially fermented teas [32]. This could explain the shift in aroma to fungal type after stuck fermentation. The formation of aroma precursor compounds from polyunsaturated fatty acids is the result of assisted oxidation by lipoxygenase (LOX) [20]. In fact, Wu et al. have isolated an *Aspergillus* strain that secretes highly active LOX [34]. In conclusion, microbial lipid metabolism during stuck fermentation may generate precursors that transform the aroma profile of FHYH.

### 3.3. Stuck Fermentation Altered the Profile of Volatile Metabolites in Summer Black Tea

The aroma of tea is a crucial component of its quality and is formed by a number of volatile metabolites. As a result, following stuck fermentation, we also examined the alterations in the volatile metabolites of summer black tea. GC-MS/MS was used to analyze six biological replicates of YH and FHYH, and the quality control evaluation revealed that the instrument and signals were dependable and stable (Appendix A). In addition, more than 75% of the substances in the QC samples had CVs less than 0.3 (Appendix A), indicating that the experimental data was stable. We detected 664 volatile metabolites, including 14 acids, 56 alcohols, 41 aldehydes, 12 amines, 41 aromatics, 117 esters, 1 ether, 4 halogenated hydrocarbons, 114 Heterocyclic compounds, 59 hydrocarbons, 51 ketones, 11 nitrogen compounds, 14 phenols, 4 sulfur compounds, 122 terpenoids, 3 other compounds (Figure 4A and Appendix A). PCA showed a clear separation between YH and FHYH based on these metabolites (Figure 4B), and HCA showed that the two samples initially separated into two clusters, while the individual replicates were grouped into one cluster (Figure 4C). In the supervised OP-LSDA model as well (Figure 4D), YH and FHYH were clearly distinct and the model was not overfitted (Appendix A). These results are indicative of significant changes in volatile metabolites after microbial stuck fermentation of summer black tea.

A total of 345 volatile metabolites were significantly different between the YH and FHYH samples (VIP ≥ 1, *p* ≤ 0.05, |Log_2_FC| ≥ 1), of which 324 were down-regulated and only 21 were up-regulated. (Figure 4E, Appendix A). KEGG enrichment analysis did not reveal enrichment of any of the pathways (Figure 4F, *p* ≤ 0.05). Based on the FlavorDB, Flavor Library, and Flavorne databases, the odor characteristics of 171 volatile metabolites were identified. Forty metabolites with fruity or sweet aromas, including 4-hexen-3-one (CAS:2497-21-4), linalyl acetate (CAS.115-95-7), hexanoic acid, 3-hexenyl ester and (Z)- (CAS:31501-11-8), were down-regulated. In addition, 21 metabolites with floral aroma, such as 1-nonanol (CAS:143-08-8), benzoic acid, 2-propenyl ester (CAS:583-04-0), 5, 9-undecadien-2-one, 6,10-dimethyl- and (Z)- (CAS:3879-26-3), also showed a decreasing trend. The down-regulation of these metabolites explains the diminished floral and sweet aromas of FHYH. Furthermore, 15 metabolites associated with a green aroma, such as heptanal (CAS:111-71-7) and (6Z)-nonen-1-ol (CAS:35854-86-5), decreased after fermentation. Qin et al. showed that metabolites with green aroma are associated with alcohols and aldehydes [35]. However, sensory evaluation indicated similar green notes in YH and FHYH tea samples, which can be attributed to the low aroma threshold metabolites 4-methylthiazole (CAS:693-95-8, aroma threshold: 0.055 mg/m^3^) and 2-isobutylthiazole (CAS:18640-74-9, aroma threshold: 0.0035 mg/m^3^). Consistent with the results reported by Zhang et al. [11], the content of acetophenone (CAS:98-86-2) increased after stuck fermentation. Acetophenone has a floral aroma and is a key determinant of the aroma characteristics of Fu brick tea. The increase in acetophenone during fermentation may be related to the moist heat effect and microbial metabolism [35]. Furthermore, acetophenone is an important contributor to fungal aromas, which consist of floral, woody, and minty notes [35]. We observed an increase in bicyclo [3.1.0] hexane and 4-methylene-1-(1-methylethyl)- (CAS:3387-41-5) that are associated with woody aroma, and in isophorone (CAS:78-59-1) that has minty notes, after stuck fermentation. Therefore, these metabolites may serve as the precursors of the fungal fragrance of FHYH. In addition, isophorone is a common product of microbial stuck fermentation, and is present at high levels in the fermented Fu brick tea, Liupao tea, Pu-erh tea, and Qing brick tea [36]. Ma et al. reported that isophorone in Liupao tea was associated with core functional microorganisms such as *Aspergillus*, *Wallemia*, *Xeromyces* and *Blastobotrys* [37]. We observed a decrease in 3-penten-2-one, 4-methyl- (CAS:141-79-7) after stuck fermentation, which is contradictory to that observed for fermented dark tea of the same species [38]. This can be attributed to the difference in the stuck fermentation substrate; while we used primary black tea, Zhang et al. used primary green tea as the substrate [38]. Furthermore, 3-penten-2-one, 4-methyl- has been detected in Keemun dark tea instead of the black or green varieties, which suggests microbial involvement [39]. Environmental factors like heat and humidity also play a significant role in the conversion of these metabolites [37]. In conclusion, stuck fermentation altered the aroma characteristics of summer black tea from floral and sweet to fungal, which may be inextricably linked to the microbial conversion of volatile metabolites.

### 3.4. Stuck Fermentation Altered the Microbial Diversity in Summer Black Tea

The changes in the microbial community of summer black tea following stuck fermentation were analyzed by 16S rRNA and ITS sequencing. The sequence quality of 16S rRNA and ITS sequencing was excellent, with Q20 greater than 99% and Q30 greater than 97% for all samples (Appendix A). Shannon and Simpson indices of bacterial (Shannon index: 5.36 to 7.86; Simpson index: 0.85 to 0.97) and fungal (Shannon index: 2.86 to 7.00; Simpson index: 0.53 to 0.97) communities were higher in FHYH compared to YH (Appendix A). The separation of the two groups of samples was evident in the principal coordination analysis (PCoA), especially that of the fungal community, indicating that the microbial community was altered by stuck fermentation (Figure 5A,F). At the phylum level, the abundance of Firmicutes, Proteobacteria, Acidobacteriota, Basidiomycota, and Mucoromycota was significantly higher in FHYH compared to YH samples (Figure 5B,G). In addition, the genera *Pullulanibacillus*, *Bacillus*, *Paenibacillus*, *Cohnella*, *Massilia*, *Aspergillus,* and *Penicillium* were more abundant in the FHYH samples (Figure 5C,H). Wang et al. found an increase in the fungal biodiversity of Tibetan tea after stuck fermentation [40], while Li et al. also concluded that there was a significant increase in the bacterial diversity of Fu brick tea at the later stages of fermentation [41], which is consistent with our research. However, both studies also reported that the microbial community diversity was not always in an increasing state throughout the microbial fermentation process, but was a dynamic change. It should also be noted that the fungal diversity of Fu brick tea decreased in the final stage of fermentation, which may be due to the fact that Fu brick tea needs to be inoculated with *Aspergillus cristatus* before fermentation, which gradually became the dominant microorganisms in the later stage, while we adopted the method of natural microbial fermentation and did not artificially add any strains of microorganisms, relying on microorganisms in the environment, which resulted in the growth and multiplication of microorganisms, and the diversity of the microorganisms increased. Of course, it is also this increase in microbial abundance that facilitates the isolation and sieving of microorganisms that potentially contribute to tea quality in our subsequent studies. In addition, we identified key microorganisms in YH and FHYH by linear discriminant analysis of effect size (LefSe), with LDA > 4 as the threshold. 16 bacterial and 10 fungal genera were screened as biomarkers in YH, including *Botryosphaeria*, *Setophoma*, *Bacteroides*, *Alloprevotella* and *Paraprevotella*, and 17 bacterial and 29 fungal genera were the key microbes in FHYH, including *Aspergillus*, *Penicillium*, *Cladophialophora*, *Cohnella*, *Paenibacillus*, *Bacillus* and *Pullulanibacillus* among others (Figure 5D,I). The evolutionary relationships between these microorganisms were assessed through a cladogram, which showed that the key microbial communities in YH were predominantly of the orders Bacteroidales, Pleosporales and Botryosphaeriales, while those in FHYH were Bacillales, Venturiales, Chaetothyriales, Eurotiales and Trechisporales (Figure 5E,J). Taken together, stuck-fermentation significantly increased the microbial richness and diversity of summer black tea.

### 3.5. Correlation Analysis of Core Microorganisms and Principal Metabolites

Spearman correlation analysis was performed on 40 metabolites (including 20 non-volatile metabolites and 20 volatile metabolites, *p* ≤ 0.05, |Log_2_FC| top 20) and 5 core genera (*Aspergillus*, *Pullululanibacillus*, *Bacillus*, *Paenibacillus*and and *Basidiomycota_gen_Incertae_sedis*, *p* ≤ 0.05, Richness ≥ 3%) to understand the relationship between microbial communities, metabolites, and tea quality.

As shown in Figure 6A, *Aspergillus*, *Pullululanibacillus,* and *Bacillus* were negatively correlated with L-histidine, L-lysine, L-norleucine, L-phenylalanine, and L-valine. Since these amino acids are associated with a bitter taste, these microbes likely play a key role in improving the taste of black tea in summer. *Aspergillus* reduces free amino acids through Maillard reaction and enzymatic breakdown [42]. Li et al. also revealed a negative correlation between L-phenylalanine and Aspergillus in green tea during stuck fermentation, and the period of stuck fermentation in this study (37 days) was similar to ours, which suggests that stuck fermentation is not only effective for black tea base in flavor improvement but may also be applicable to other tea types [18]. In fact, *Aspergillus* is known to secrete multiple enzymes, including α-amylases, glucoamylase, various cellulases, pectinase, xylanase and hemicellulose, proteases, and polyphenol oxidase [42]. *A. niger* reduces acidity, bitterness, and astringency in unfermented tea through extracellular enzyme action, and promotes the sweet and mellow quality of fermented tea [42]. *Paenibacillus*also secretes a variety of enzymes, including amylase, cellulase, hemicellulase, lipase, pectinase, and lignin-modifying enzymes [43]. Xiang et al. found that *Bacillus* can oxidize tea catechins into theaflavins by producing pectinase and polyphenol oxidase, which reduces bitterness and brightens the color of tea soup [44]. The breakdown of cellulose and pectin increases the sugar content of tea soup, which is consistent with the significant positive correlation of *Penicillium*, *Basidiomycota_gen_Incertae_sedis*, *Pullulanibacillus* and *Bacillus* with various sugars (D-mannitol (CAS:69-65-8), D-sorbitol (CAS:50-70-4), dulcitol (CAS:608-66-2)). Wang et al. showed that the surface of tea leaves is covered with microorganisms during fermentation, which eventually disrupts the leaf cell structure [33]. *Penicillium* also degrades cellulose to produce a small amount of free sugar, which lends sweetness to the tea soup. In addition, its secondary metabolite penicillin inhibits the growth of harmful bacteria [42]. These results suggest that microorganisms contribute significantly to the degradation of tea leaf cell walls. *Basidiomycota_gen_Incertae_sedis*, a genus of phylum Basidiomycota, may be involved in the production of various metabolites (D-mannitol, D-sorbitol, dulcitol, guanine (CAS:73-40-5), 5-hydroxymethylfurfural (CAS:67-47-0), 3-methyl-2-oxobutanoic acid (CAS:759-05-7) in tea, although only a few studies have reported on this species. Overall, our findings suggest that microbial activity may play a significant role in metabolite conversion, reduced bitterness, and increased sweetness in stuck-fermented summer black tea.

As shown in Figure 6B, *Aspergillus* exhibited a strong negative correlation with 12 metabolites, including p-menth-8-en-3-ol, acetate (CAS:89-49-6), linalyl acetate (CAS:115-95-7), etc.) and strong positive correlations with 2 metabolites (cyclohexanecarboxylic acid, methyl ester (CAS:4630-82-4) and 5-methyl-(E)-2-hepten-4-one (CAS:102322-83-8). Five of these metabolites are esters and impart a fruity aroma. The extracellular hydrolases secreted by *Aspergillus* may have converted these esters to other aromatic substances, thereby altering the aroma profile of summer black tea from fruit-like sweet and floral to fungal during stuck fermentation. *Aspergillus* is often introduced in fermented foods to produce various metabolites [35]. For instance, it is the core microorganism in Pu-erh tea and catalyzes the production of linalool and its oxides during fermentation [35]. In addition, the crude enzymes secreted by *Aspergillus* regulate the aroma of traditional green tea or other tea products [35]. Qin et al. demonstrated that *Aspergillus* is the dominant fungus in Fu brick tea, and its metabolic activity enhances the fungal aroma [35]. The genus *Bacillus* was negatively correlated with 1-nonanol (CAS:143-08-8) and positively correlated with 5-methyl-(E)-2-hepten-4-one. *Bacillus bruxellensis* and *Acetobacter indonesiensis* are known to oxidize 1-nonanol to nonanoic acid [45]. *Bacillus* also produces volatile substances, including pyrazines, aldehydes, ketones, and alcohols, which are the precursors of aromatic compounds in some alkaline fermented foods, condiments, and wines [46]. *Penicillium multicolor* synthesizes the aroma precursor substance β-primeverosides [35]. *Paenibacillus*is also a major microorganism in some traditionally fermented foods [44]. However, little is known regarding the impact of *Penicillium* and *Paenibacillus* on the production of volatile metabolites during tea fermentation. *Basidiomycota_gen_Incertae_sedis* is an unknown microorganism that likely plays a key role in shaping the metabolite profile of fermented tea, and warrants further investigation. In summary, microbes alter the quality, taste, and aroma of summer black tea during stuck fermentation by producing extracellular enzymes.

## 4. Conclusions

In the current study, a method to improve the quality of summer tea using stuck fermentation was proposed, and we investigated the tea quality, metabolic profile, and microbial community of summer black tea after stuck fermentation in detail, and explored the relationship between tea quality and metabolic profile and microbial community. Stuck fermentation significantly improved the quality of summer black tea by reducing the bitterness, astringency, sourness, and freshness, enhancing the sweetness, mellowness, and smoothness, and altering the aroma from sweet and floral to fungal. Flavonoids, phenolic acids, amino acids, and their derivatives that lend a bitter and astringent taste were down-regulated. In addition, organic acids, alkaloids with a sour and bitter taste, and glycoconjugates of amino acids and flavonoids that release glycosides were also significantly reduced after stuck fermentation. In contrast, sweet flavor saccharides and lipid precursors of aroma substances increased significantly. Stuck fermentation also led to the downregulation of floral and sweet aroma-related metabolites and increased the levels of acetophenone and isophorone, which contribute to fungal aroma. Furthermore, the richness of fungal and bacterial communities in summer black tea increased significantly after stuck fermentation, and *Aspergillus*, *Pullululanibacillus*, *Bacillus*, *Paenibacillus*, *Basidiomycota_gen_Incertae_sedis* were identified as the core microorganisms. Finally, the improvement of tea quality and the transformation of metabolites during stuck fermentation were closely related to microbial activities, especially the secretion of extracellular enzymes. Taken together, stuck fermentation significantly improved the quality of summer black tea by altering its metabolic profile and microbial composition. Our research has searched for a valuable way to improve the quality of summer tea, alleviate the waste of resources caused by tea farmers not picking summer tea, put forward an idea for the enhancement of the utilization of summer tea, and provide a reference for flavor improvement and product innovation in the food industry. Although stuck fermentation improves the quality of summer tea, there are some unknown strains such as *Basidiomycota_gen_ Incertae_sedis* in the fermentation; therefore, isolation of strains that contribute to the improvement of summer tea quality and functional validation of these strains need further study.

## Figures and Tables

**Figure 1 foods-12-03414-f001:**
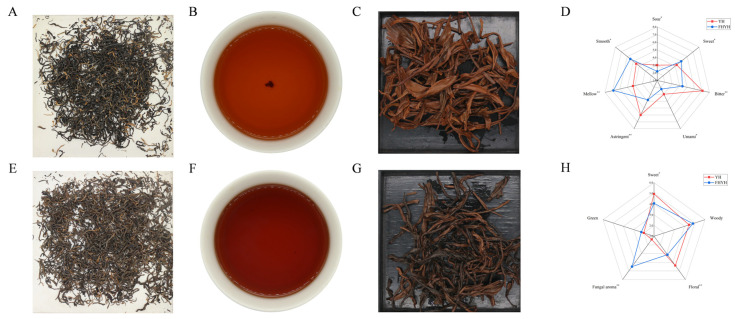
Sensory evaluation and electronic tongue analysis. (**A**) Dry tea of YH. (**B**) Tea soup of YH. (**C**) Tea base of YH. (**D**) Quantitative descriptive analysis of taste (*: *p* < 0.05, **: *p* < 0.01). (**E**) Dry tea of FHYH. (**F**) Tea soup of FHYH. (**G**) Tea base of FHYH. (**H**) Quantitative descriptive analysis of aroma (*: *p* < 0.05, **: *p* < 0.01).

**Figure 2 foods-12-03414-f002:**
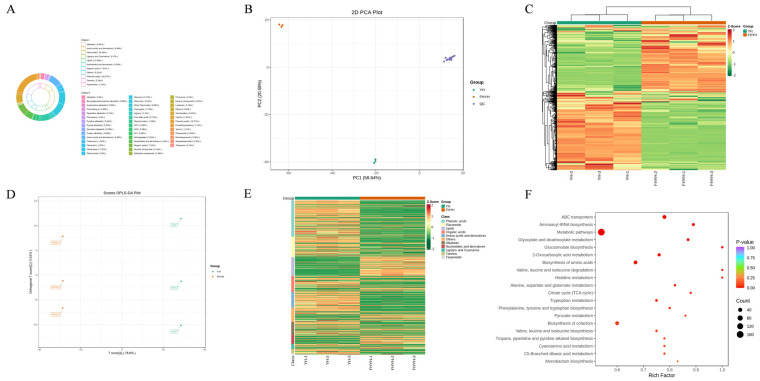
Analysis of non-volatile metabolites in YH and FHYH groups. (**A**) The categories of 1248 non-volatile metabolites. (**B**) Principal component analysis (PCA) of YH and FHYH groups. PC1 is the first principal component. PC2 is the second principal component. (**C**) Hierarchical clustering analysis (HCA) of 1248 non-volatile metabolites. (**D**) Orthonormal partial least squares discriminant analysis (OPLS-DA) of YH and FHYH. (**E**) Heat map of differential metabolites. (**F**) KEGG pathway enrichment analysis of significantly different non-volatile metabolites.

**Figure 3 foods-12-03414-f003:**
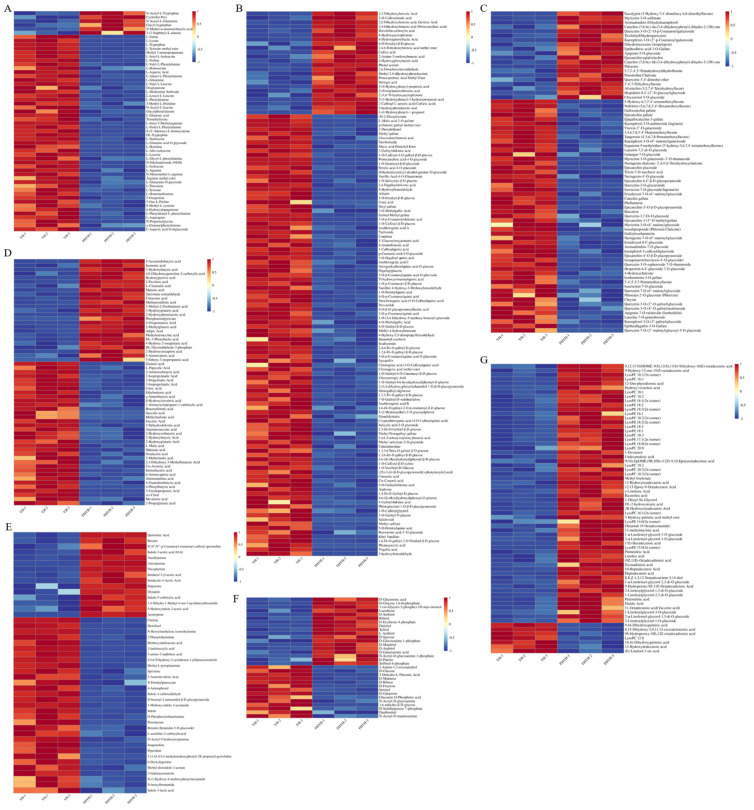
Heatmap of significantly different non-volatile metabolites for each class. (**A**) Amino acids and derivatives. (**B**) Phenolic acids. (**C**) Flavonoids. (**D**) Organic acids. (**E**) Alkaloids. (**F**) Saccharides. (**G**) Lipids.

**Figure 4 foods-12-03414-f004:**
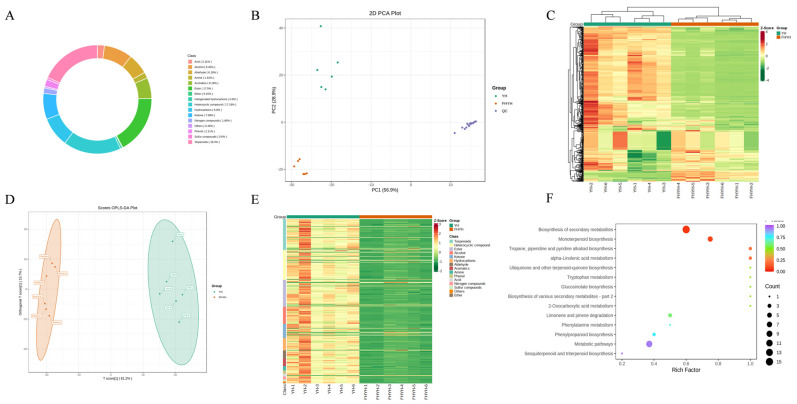
GC-MS/MS analysis of YH and FHYH groups. (**A**) The categories of 664 volatile metabolites. (**B**) Principal component analysis (PCA) of YH and FHYH. PC1 is the first principal component. PC2 is the second principal component. (**C**) Hierarchical clustering analysis (HCA) of 664 volatile metabolites. (**D**) Orthonormal partial least squares discriminant analysis (OPLS-DA) of YH and FHYH. (**E**) Heat map of differential metabolites. (**F**) KEGG pathway enrichment analysis of significantly different non-volatile metabolites.

**Figure 5 foods-12-03414-f005:**
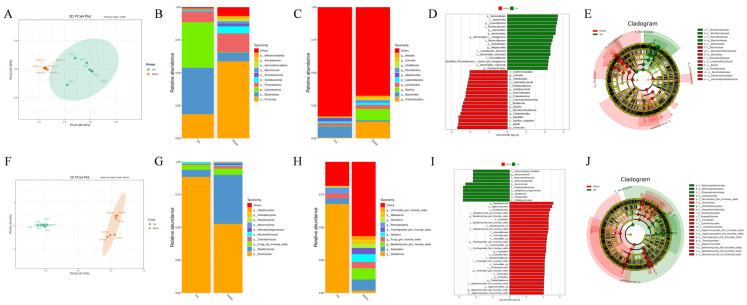
Diversity and composition of the microbial community in YH and FHYH. (**A**) Principal coordination analysis (PCoA) for bacteria. (**B**) Relative abundance of bacteria at the phylum level. (**C**) Relative abundance of bacteria at the genus level. (**D**) The linear discriminant analysis effect size (LefSe) of LDA score distribution histogram for bacteria. (**E**) LEfSe analysis of an evolutionary branching diagram for bacteria. (**F**) PCoA for fungi. (**G**) Relative abundance of fungi at the phylum level. (**H**) Relative abundance of fungi at the genus level. (**I**) LefSe of LDA score distribution histogram for fungi. (**J**) LEfSe analysis of an evolutionary branching diagram for fungi.

**Figure 6 foods-12-03414-f006:**
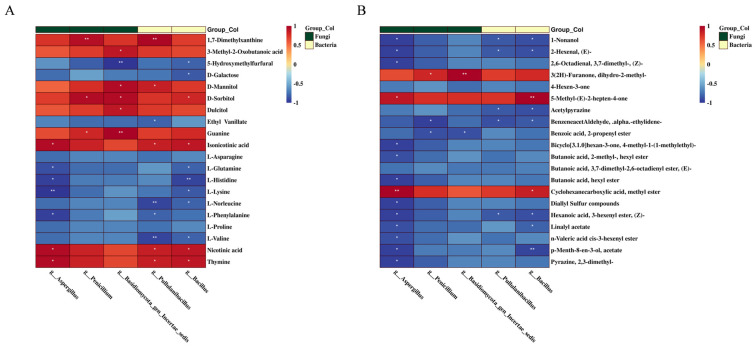
Spearman correlation analysis of major differential metabolites and key microorganisms. (**A**) Non-volatile metabolites. (**B**) Volatile metabolites. * *p* < 0.05 and ** *p* < 0.01.

## Data Availability

The data used to support the findings of this study can be made available by the corresponding author upon request.

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
