# Peer review of "Metabolome and Microbiome Analysis to Study the Flavor of Summer Black Tea Improved by Stuck Fermentation"

_foods, 2023, doi:10.3390/foods12183414_

Round 1
Reviewer 1 Report
Comments and Suggestions for Authors
The presented research article on "Stack-fermentation improved the flavor of summer black tea by altering the microbial and metabolic composition" provides an overview of how microbial fermentation can influence the quality of summer black tea. However, to provide constructive feedback and suggest potential points of improvement, I have broken down the article into various sections:
1. Title and Abstract:
· The title is straightforward and gives a clear idea about the focus of the study. However, it could be more concise.
· The abstract summarizes the study well, but clarity on the specific microbes responsible for the positive changes might be beneficial.
2. Introduction:
· The introduction lays out the context and importance of the study but can delve deeper into the scientific background of microbial fermentation in tea to establish a stronger foundation for the study.
3. Methods:
· Experimental materials: While the source of tea leaves is mentioned, any potential bias or limitation due to this specific source should be highlighted.
· Tea preparation: The procedure might benefit from a flowchart or diagram for better visualization.
· Sensory evaluation: Clarification on how professionals were selected and trained can improve the reliability of the study.
· Analysis methods (UPLC-MS/MS, GC-MS/MS, etc.): These are well-detailed. Including reasons for specific methodological choices might be helpful for readers unfamiliar with these techniques.
4. Results and Discussion
· Sensory Evaluation: The flavor and sensory characteristics of the tea samples were determined through traditional sensory evaluation. This subjective method might introduce bias as it relies on individual perceptions.
· Color Assessment: Color descriptions like "yellowish auburn", "clear orange-red", or "deep red and dull" can be subjective and can vary depending on the observer.
· Reference Comparisons: The study draws comparisons with other findings (e.g., Xiao et al.) without providing direct contrasts in experimental conditions.
· Fungal Aroma Subjectivity: The aroma description of "fungal" might be subjective as it is not a universally recognized descriptor and can vary among individuals.
· Limited Microbial Examination: While the microbial communities identified are detailed, the study may not account for all potential microbes or the interactions between these microbes.
· Predominance on Certain Microbes: The study places a lot of emphasis on some specific microbes like Aspergillus, potentially overlooking the effects of lesser-known microbes.
· In-depth Mechanistic Insights: The exact biochemical mechanisms by which microbes transform the taste and aroma profiles of tea aren't fully elucidated.
· External Validity: The study's findings might be specific to the particular type of tea or the specific environmental conditions under which the fermentation took place.
· Lack of Replication Details: The study doesn't mention how many times the experiments were repeated, which could raise concerns about reproducibility.
· Unknown Microorganisms: Some microorganisms, like Basidiomycota_gen_Incertae_sedis, are indicated as unknown, suggesting that there's a lack of comprehensive knowledge about their roles.
· Limitations of Spearman Correlation: The use of Spearman correlation is solely based on rank and doesn't capture potential non-linear relationships between metabolites and microorganisms.
· Single Method of Fermentation: Only stack-fermentation is explored, which doesn't allow for comparisons with other fermentation techniques that might also improve tea quality.
· Focus on Summer Black Tea: Findings might not be generalizable to black teas produced in other seasons or other types of teas.
· Causal Relationships: While correlations are drawn between microbial presence and tea quality, direct causal relationships might not be firmly established.
5. General Comments:
· Formatting: The article might benefit from more subheadings and clear delineation between methods to improve readability.
· Figures and Tables: Assuming these are part of the article, clear, concise, and informative captions are vital. Including more visual aids like charts or graphs to represent key findings might be beneficial.
· References: Ensure all references are up-to-date, relevant, and cited correctly. Providing some recent studies could offer readers the latest context.
· Conclusion: Summarize key findings and their implications succinctly. Discuss potential applications of this research in the industry.
· Acknowledgments & Conflicts of Interest: It's crucial to acknowledge any funding sources and declare any potential conflicts of interest to maintain transparency.
6. Potential Research Extensions:
· Consider discussing how stack-fermentation compares to other fermentation methods.
· Delve deeper into the economic benefits of improving the quality of summer tea through fermentation.
Overall, the article provides a detailed insight into the impact of microbial fermentation on the quality of summer black tea. However, refining the content, focusing on clarity and readability, and expanding on some areas can make it more comprehensive and user-friendly.
Comments on the Quality of English Language
Minor editing of English language required
Author Response
Dear Reviewer,
Thank you very much for your careful review and constructive suggestion with regard to our manuscript “Stack-fermentation improved the flavor of summer black tea by altering the microbial and metabolic composition” (ID: foods-2594288). Those comments are all valuable and very helpful for revising and improving our paper, as well as the important guiding significance to our researches. We have studied comments carefully and made correction which we hope meet with approval. Revised portion are marked in red in the paper. The main corrections in the paper and the responds to the reviewer’s are as follows:
Revisions in accordance with Reviewer 1’s comments
1. The title is straightforward and gives a clear idea about the focus of the study. However, it could be more concise.
Response and Revision: Thanks for your valuable suggestions. I have revised the title to “Metabolome and microbiome based to study flavor of summer black tea improved by stuck-fermentation”.
2. The abstract summarizes the study well, but clarity on the specific microbes
Response and Revision: Thanks for your valuable suggestions. I have added the specific microbes responsible for the positive changes in “Abstract”.
3. The introduction lays out the context and importance of the study but can delve deeper into the scientific background of microbial fermentation in tea to establish a stronger foundation for the study.
Response and Revision: Thanks for your valuable suggestions. I have added the scientific background of microbial fermentation of tea in the “Introduction”.
4. Experimental materials: While the source of tea leaves is mentioned, any potential bias or limitation due to this specific source should be highlighted.
Response and Revision: Thanks for your valuable suggestions. I have added the restrictiveness of the source of the tea in “2.2. Tea preparation”.
5. Tea preparation: The procedure might benefit from a flowchart or diagram for better visualization.
Response and Revision: Thanks for your valuable suggestions. I have added a diagram of the tea preparation in “2.2. Tea preparation”.
6. Sensory evaluation: Clarification on how professionals were selected and trained can improve the reliability of the study.
Response and Revision: Thanks for your valuable suggestions. In order to make the results representative and reproducible, the professionals were controlled to be one male and one female aged 20-30, 30-40, 40-50 and 50-60, and all 8 professionals passed the qualification examination for tea evaluation. One week before the formal sensory evaluation, the 8 professionals further practiced the rules, terminology, and process of black tea and microbial fermented tea evaluation and prohibited spicydiets.
7. Analysis methods (UPLC-MS/MS, GC-MS/MS, etc.): These are well-detailed. Including reasons for specific methodological choices might be helpful for readers unfamiliar with these techniques.
Response and Revision:Thanks for your valuable suggestions. I have added the reasons for the selection of specific methods to the “Introduction”.
8. Sensory Evaluation: The flavor and sensory characteristics of the tea samples were determined through traditional sensory evaluation. This subjective method might introduce bias as it relies on individual perceptions.
Response and Revision: Thanks for your valuable suggestions. I agree with you that traditional sensory evaluation relies to some extent on subjective judgment. However, sensory evaluation has always been an important means of assessing the flavor of a product within the tea or food industry. Secondly, as a food product, the most direct way for people to judge the quality of tea in their daily life is to taste it. In addition, in order to minimize the influence of personal subjective factors, age and gender were taken into account when selecting professionals, and the evaluation was conducted blindly, without any one person knowing what treatments had been done to the two samples.
9. Color Assessment: Color descriptions like "yellowish auburn", "clear orange-red", or "deep red and dull" can be subjective and can vary depending on the observer.
Response and Revision: Thanks for your valuable suggestions. I agree with you about the color of the tea soup being dependent on the observer. In order to minimize errors, we select professional reviewers considering age and gender, and use uniform light intensity during observation, and require observation from different angles, and in order to avoid visual fatigue, we do not allow prolonged (<30s) observation of the same tea soup.
10. Reference Comparisons: The study draws comparisons with other findings (e.g., Xiao et al.) without providing direct contrasts in experimental conditions.
Response and Revision: Thanks for your valuable suggestions. I have added a comparison of experimental conditions in “3.1. Stack fermentation improved the sensory quality of summer black tea” and “3.5. Correlation analysis of core microorganisms and principal metabolites”.
11. Fungal Aroma Subjectivity: The aroma description of "fungal" might be subjective as it is not a universally recognized descriptor and can vary among individuals.
Response and Revision: Thanks for your valuable suggestions. In the Chinese standard (GB/T 14487-2017), fungal aroma refers to a special aroma unique to microbial fermented teas (e.g., Fu brick tea). In order for the professional reviewers to accurately determine this aroma, we required the professional reviewers to practise the rules, processes and terminology of the review of black teas and microbial fermented teas one week prior to the formal sensory evaluation, and selected a number of microbial fermented teas (Fu brick tea, Liubao tea, Qing brick tea and Pu-er Tea) for the purpose of recognizing the type of aroma.
12. Limited Microbial Examination: While the microbial communities identified are detailed, the study may not account for all potential microbes or the interactions between these microbes.
Response and Revision: Thanks for your valuable suggestions. I couldn't agree with you more, but our current research can't show all the microbial interactions, and we will look into it further when experimental conditions allow.
13. Predominance on Certain Microbes: The study places a lot of emphasis on some specific microbes like Aspergillus, potentially overlooking the effects of lesser-known microbes.
Response and Revision:Thanks for your valuable suggestions. I agree with you that Aspergillus was discussed more in this study because it is currently one of the most studied species in the field of fermented tea, and changes in its flora are closely related to the quality of tea. In addition, Aspergillus was one of the most significantly varied species at the genus level in our study, and the LefSe analysis also showed that Aspergillus was a biomarker in FHYH, which are suggesting that Aspergillus contributes positively to the flavor improvement of summer black tea. However, it cannot be denied that the flavor improvement of black tea in summer is the result of all microorganisms working together, and there are even some unknown microorganisms. Based on the current experimental data, we lack direct evidence for the role of microorganisms to a certain extent, and need to speculate with the help of previous studies and the current results. Therefore, our further research will focus on the identification of unknown microorganisms and the isolation of some lesser-known microorganisms for validation.
14. In-depth Mechanistic Insights: The exact biochemical mechanisms by which microbes transform the taste and aroma profiles of tea aren't fully elucidated.
Response and Revision: Thanks for your valuable suggestions. I strongly agree with you, and we will follow up with further research on the specific mechanisms involved.
15. External Validity: The study's findings might be specific to the particular type of tea or the specific environmental conditions under which the fermentation took place.
Response and Revision: Thanks for your valuable suggestions. Our current results are indeed the result of specific tea plant varieties or fermentation conditions, and we will conduct replicated trials to validate the current results and guide tea companies to utilize Yinghong No. 9 for microbial fermentation of summer black teas to meet the diversified needs of consumers for tea.
16. Lack of Replication Details: The study doesn't mention how many times the experiments were repeated, which could raise concerns about reproducibility.
Response and Revision: Thanks for your valuable suggestions. I have added the biological replicates of the sample preparation in “2.2. Tea preparation” and described the number of replicates of the data in detail in “2.7. Statistical analysis”.
17. Unknown Microorganisms: Some microorganisms, like Basidiomycota_gen_Incertae_sedis, are indicated as unknown, suggesting that there's a lack of comprehensive knowledge about their roles.
Response and Revision: Thanks for your valuable suggestions. Because the fermentation method used in this study is the natural accumulation fermentation of microorganisms, many microorganisms in the environment will participate in it, and inevitably there are some under-recognized strains, and even the emergence of unknown microorganisms, which is the next step of our research, and we need to isolate and identify the microorganisms that are unknown to us, as well as the verification of the function of these microorganisms.
18. Limitations of Spearman Correlation: The use of Spearman correlation is solely based on rank and doesn't capture potential non-linear relationships between metabolites and microorganisms.
Response and Revision: Thanks for your valuable suggestions. I agree with you, but the main focus in this research was on the linear relationship between metabolites and microorganisms, and in subsequent research I will emphasize the non-linear relationship between the two.
19. Single Method of Fermentation: Only stack-fermentation is explored, which doesn't allow for comparisons with other fermentation techniques that might also improve tea quality.
Response and Revision: Thanks for your valuable suggestions. Our study mainly explored the processing of black tea into microbiologically fermented black tea using stuck-fermentation, so no comparison with other methods was made for the time being.
20. Focus on Summer Black Tea: Findings might not be generalizable to black teas produced in other seasons or other types of teas.
Response and Revision: Thanks for your valuable suggestions.Tea according to the season is divided into spring tea, summer tea, autumn tea, of which summer tea because of high polyphenol, low amino acid and other characteristics lead to the production of the lowest quality of tea, so many places will not directly do summer tea. Of course, if summer tea must be used, it is more appropriate to make black tea, so our research is centered on summer black tea. In addition, the use of microbial fermentation to improve tea quality is also feasible in other seasons or types of tea. For example, other reports have found that microbial fermentation can improve the quality of autumn green tea (https://doi.org/10.1016/j.foodchem.2021.129848).
21. Causal Relationships: While correlations are drawn between microbial presence and tea quality, direct causal relationships might not be firmly established.
Response and Revision: Thanks for your valuable suggestions. I strongly agree with you.Our study is more speculative based on the existing results and combined with related literature reports, due to the method of natural microbial fermentation, involving a large number of strains, the existing experimental conditions do not allow to get the direct evidence, subsequent to the isolation and identification of microorganisms that play a potential role, and further study of their functions, in order to provide direct evidence for the relationship between microbial fermentation and the quality of tea.
22. The article might benefit from more subheadings and clear delineation between methods to improve readability.
Response and Revision: Thanks for your valuable suggestions. In order to improve the readability of the article, I added subheadings, including “Introduction”, “Materials and Methods”, “Results and discussion”, and “Conclusions”. In the "Materials and Methods" section, I have described in detail the preparation of the samples, as well as various analytical methods and procedures, including traditional sensory evaluation, LC-MS/MS, GC-MS/MS, 16S rRNA and ITS sequencing.
23. Figures and Tables: Assuming these are part of the article, clear, concise, and informative captions are vital. Including more visual aids like charts or graphs to represent key findings might be beneficial.
Response and Revision: Thanks for your valuable suggestions. Figures and graphs are important elements of the article, and I provided the editorial board with clear original figures and raw data, and annotated the contents of the figures and graphs using clear and concise headings.
24. References: Ensure all references are up-to-date, relevant, and cited correctly. Providing some recent studies could offer readers the latest context.
Response and Revision: Thanks for your valuable suggestions. I have revisited the references throughout the text again to ensure that the literature cited is relevant, up-to-date and correctly cited, which also provides a suitable research context for the study in question.
25. Conclusion: Summarize key findings and their implications succinctly. Discuss potential applications of this research in the industry.
Response and Revision: Thanks for your valuable suggestions. I have distilled the key conclusions of the whole manuscript in the "Conclusions" and proposed applications of our research results, as well as indicating directions for further research.
26. Acknowledgments & Conflicts of Interest: It's crucial to acknowledge any funding sources and declare any potential conflicts of interest to maintain transparency.
Response and Revision: Thanks for your valuable suggestions. I have acknowledged all funding sources in "Acknowledgments" and declared that we have no have no known competing financial interests or personal relationships in "Declaration of competing interest".
- Consider discussing how stack-fermentation compares to other fermentation methods.
Response and Revision: Thanks for your valuable suggestions. Our research mainly explores the processing of black tea into microbiologically fermented black tea by using stuck-fermentation, so more comparisons with microbiologically fermented tea are cited throughout the article. In addition, another purpose of this study is to screen microorganisms with potential benefits for improving tea quality, which will be subsequently isolated and characterized, and the ability to improve tea quality will be verified, as well as comparative studies with other methods.
- Delve deeper into the economic benefits of improving the quality of summer tea through fermentation.
Response and Revision: Thanks for your valuable suggestions. I have added the relevant content in “4. Conclusions”. The black tea made from summer tea fresh has a heavy bitter and astringent flavor and poor quality, which leads to many tea farmers are reluctant to pick, resulting in a serious waste of summer tea resources. We made black tea from summer tea fresh of Yinghong No. 9 and used microorganisms to ferment it, which reduced the bitterness and astringency of the black tea and improved the quality of the tea, which provided a reference to alleviate the wastage of summer tea resources and helped to increase the economic income of tea farmers.

Reviewer 2 Report
Comments and Suggestions for Authors
Authors studied the organoleptic characteristics, the metabolic activity and the microbiota profile of stuck-fermented summer black tea. Results indicated that fermentation process altered flavor and aroma characteristics of the product mainly due to the microbial activity of specific fungal and bacterial species which in turn led to the production of a specific metabolic profile. Authors suggest that this is an appropriate method for the improvement of overall quality of black tea. In general, the work is interesting and the aim is clear. The manuscript is generally well-written and shows significant importance. However, there are some issues that have to be carefully addressed by authors.
-L2. I think that it is “stuck” fermentation. Please revise throughout the manuscript.
-L31. Please revise.
-L35-36. What kind of health benefits? Please specify.
-L176-178. What about bioinformatic analysis? Please provide a thorough description.
-L181-183. More details are needed herein e.g., which rotation, distance etc. was used for PCA and HCA, how data were transformed and scaled, etc.
-L227-236. A more thorough discussion is needed herein.
-L478-479. What about sequences quality obtained, and how many of them were retained. Please provide more information about those aspects.
-L484-494. A deeper discussion is needed about the findings of microbial composition in FHYH, as well as about the biomarkers suggested by the authors.
-L598-600. A final paragraph (either herein or at the end of discussion section) clearly highlighting the importance of this work, how industry could be benefited by those findings, as well as what authors suggest as future perspectives, is strongly recommended.
Comments on the Quality of English LanguageMinor editing of English language is suggested
Author Response
Dear Reviewer,
Thank you very much for your careful review and constructive suggestion with regard to our manuscript “Stack-fermentation improved the flavor of summer black tea by altering the microbial and metabolic composition” (ID: foods-2594288). Those comments are all valuable and very helpful for revising and improving our paper, as well as the important guiding significance to our researches. We have studied comments carefully and made correction which we hope meet with approval. Revised portion are marked in red in the paper. The main corrections in the paper and the responds to the reviewer’s are as follows:
Revisions in accordance with Reviewer 2’s comments
1. -L2. I think that it is “stuck” fermentation. Please revise throughout the manuscript.
Response and Revision: Thanks for your valuable suggestions. I have revised “stack” to “stuck” throughout the manuscript.
2. -L31. Please revise.
Response and Revision: Thanks for your valuable suggestions. I have revised it in the “Abstract”
3. -L35-36. What kind of health benefits? Please specify.
Response and Revision: Thanks for your valuable suggestions.I have detailed the potential health benefits of tea in the “Introduction”.
4. -L176-178. What about bioinformatic analysis? Please provide a thorough description.
Response and Revision: Thanks for your valuable suggestions.I have added bioinformatic analysis in “2.6. Analysis of microbial community by 16S rRNA and ITS sequencing”.
5. -L181-183. More details are needed herein e.g., which rotation, distance etc. was used for PCA and HCA, how data were transformed and scaled, etc.
Response and Revision: Thanks for your valuable suggestions. I have added specific details of HCA and PCA in “2.7. Statistical analysis”.
6. -L227-236. A more thorough discussion is needed herein.
Response and Revision: Thanks for your valuable suggestions. I have added a deeper discussion in “3.2. Stack fermentation altered the profile of non-volatile metabolites in summer black tea”.
7. -L478-479. What about sequences quality obtained, and how many of them were retained. Please provide more information about those aspects.
Response and Revision: Thanks for your valuable suggestions. I have added the sequences quality of 16S rRNA and ITS sequencing in "3.4. Stuck-fermentation altered the microbial diversity in summer black tea".
8. -L484-494. A deeper discussion is needed about the findings of microbial composition in FHYH, as well as about the biomarkers suggested by the authors.
Response and Revision: Thanks for your valuable suggestions. I have added a deeper discussion in “3.4. Stack-fermentation altered the microbial diversity in summer black tea”, where the section on biomarkers is expanded in “3.5. Correlation analysis of core microorganisms and principal metabolites”.
9. -L598-600. A final paragraph (either herein or at the end of discussion section) clearly highlighting the importance of this work, how industry could be benefited by those findings, as well as what authors suggest as future perspectives, is strongly recommended.
Response and Revision: Thanks for your valuable suggestions.I have already emphasized the importance of our work and the direction of further research in the “Conclusions”.
